# The characteristics and clinical course of patients with melioidosis and cancer

Tej Shukla[1], Simon Smith[1], Kristoffer Johnstone[2], Patrick Donald[1], Josh Hanson [1,3] *

**1** Department of Medicine, Cairns Hospital, Cairns, Queensland, Australia, **2** Pharmacy department, Cairns Hospital, Cairns, Queensland, Australia, **3** The Kirby Institute, University of New South Wales, Sydney, New South Wales, Australia

* jhanson@kirby.unsw.edu.au

**Data Availability Statement:** Data cannot be shared publicly because of the Queensland Public Health Act 2005. Data are available from the Far North Queensland Human Research Ethics Committee (contact via email FNQ_HREC@health.

## Abstract

### Background

Patients with an active cancer are more likely to develop melioidosis, but the characteristics and clinical course of melioidosis in patients with cancer have not been examined in detail. Trimethoprim/sulfamethoxazole (TMP-SMX) prophylaxis is prescribed to prevent melioidosis in patients receiving immune suppressing anti-cancer therapy in some jurisdictions–and is recommended in national Australian guidelines–however the risks and benefits of this strategy are incompletely defined.

### Methods

The study took place in Far North Queensland (FNQ) in tropical Australia. The characteristics and clinical course of patients with melioidosis diagnosed in the FNQ region between January 1, 1998, and June 1, 2023, who had–and did not have–an active cancer were compared. We also determined the subsequent incidence of melioidosis in patients receiving immune suppressing anti-cancer therapy in the FNQ region between January 1, 2008, and June 1, 2023, who did–and did not–receive TMP-SMX chemoprophylaxis for *Pneumocystis jirovecii* infection.

### Results

An active cancer was present in 47/446 (11%) cases of melioidosis diagnosed between January 1, 1998, and June 1, 2023; there was no association between melioidosis and any cancer type. Patients with melioidosis and cancer were more likely to be older (odds ratio (OR) (95% confidence interval (CI): 1.05 (1.03–1.08) P<0.0001) and immunosuppressed (OR (95% CI): 11.54 (5.41–24.6), p<0.0001) than patients without cancer. Immune suppressing anti-cancer therapy had been prescribed to 17/47 (36%) in the 12 months prior to their diagnosis of melioidosis. Only 10/47 (21%) with cancer and melioidosis in the cohort had received no immune suppressing anti-cancer therapy and had no other risk factors for melioidosis. Twelve months after the diagnosis of melioidosis, 25/47 (53%) were still alive; 9/22 (41%) deaths were due to melioidosis and 13/22 (59%) were due to the underlying cancer.

qld.gov.au) for researchers who meet the criteria for access to confidential data.

**Funding:** The author(s) received no specific funding for this work.

**Competing interests:** The authors have declared that no competing interests exist.

Between 2008 and June 2023, there were 4400 individuals who received myelosuppressive anti-cancer therapy in the FNQ region. There was no significant difference in the incidence of melioidosis between patients who did–and did not–receive TMP-SMX chemoprophylaxis with their myelosuppressive anti-cancer therapy (1/737 (0.15%) versus 16/3663 (0.44%); relative risk (95% confidence interval): 0.31 (0.04–2.34), p = 0.20) and no significant difference in the incidence of fatal melioidosis (0/737 versus 3/3663 (0.08%), p = 0.58).

## Conclusions

Patients with cancer are predisposed to developing melioidosis and immune suppressing anti-cancer therapy increases this risk further. However, in this region of Australia, there was no significant difference in the subsequent development of melioidosis in patients who did–and did not–receive TMP-SMX chemoprophylaxis during their myelosuppressive anti-cancer therapy.

## Author summary

In this study from a tertiary hospital in tropical Australia, an active cancer was present in 11% of 446 patients with cultured-confirmed melioidosis, However, no solid organ or haematological cancer diagnosis appeared to be associated with developing the infection. Approximately half of the individuals with cancer who developed melioidosis died within 12 months of their melioidosis diagnosis, however, deaths were more commonly due to their underlying cancer than their melioidosis. Immune suppressing anti-cancer therapy had been prescribed in the prior 12 months in 36% of the individuals with cancer and melioidosis. Trimethoprim-sulfamethoxazole (TMP-SMX) is not prescribed routinely to prevent melioidosis in local patients receiving myelosuppressive anti-cancer therapy, but individuals receiving twice weekly TMP-SMX chemoprophylaxis against *Pneumocystis jirovecii* pneumonia did not have a statistically lower rate of melioidosis than individuals who did not receive this chemoprophylaxis. Melioidosis was a rare complication of myelosuppressive anti-cancer therapy in the region. Only 16/3663 (0.44%) individuals not receiving TMP-SMX chemoprophylaxis developed melioidosis, and only 3/3663 died from the disease (0.08%). As TMP-SMX has a range of potential side effects–and has the potential to interrupt the delivery of anti-cancer therapy and drive resistance to an important antibiotic–our data do not presently support TMP-SMX prophylaxis for melioidosis in patients receiving anti-cancer therapies in this region of Australia.

## Introduction

Melioidosis is a disease caused by *Burkholderia pseudomallei*, a gram-negative bacterium found in the soil and surface water of tropical and subtropical regions [1]. Statistical modelling of clinical cases and environmental studies estimated that in 2015 there may have been 165000 cases of melioidosis globally and 89000 deaths [2]. With greater environmental disruption from urban expansion, a growing burden of health conditions that increase the risk of melioidosis and, potentially, the impact of climate change, the global burden of melioidosis is

anticipated to rise [1,3–5]. Even in well-resourced settings the case fatality rate of melioidosis approaches 10% [6].

Melioidosis usually occurs in individuals with specific comorbidities that include diabetes mellitus, hazardous alcohol consumption, chronic lung disease, chronic kidney disease, and immunosuppression [1]. Patients with cancer are also more likely to develop melioidosis, a result of the immunosuppressive anti-cancer therapy that they frequently receive, although other factors including medical comorbidities and poor nutritional status also contribute to their greater risk [7]. Despite the higher incidence of melioidosis in patients with cancer, the characteristics and clinical course of melioidosis in these patients have rarely been described in detail [1,8,9].

But it is important to define the presentation and optimal clinical management of patients with melioidosis and cancer because cancer incidence is expected to increase globally in the next 20 years, particularly in the transitioning countries which bear a disproportionate burden of melioidosis and where the number of cases of cancer-related melioidosis would be expected to rise [10].

As the case-fatality rate of melioidosis can rise to 50% in these countries, there is also interest in developing strategies that might reduce the incidence of the disease [1,11]. Public health interventions to prevent melioidosis have been unsuccessful and there is no effective vaccine, but chemoprophylaxis has shown utility in some high-risk populations [1]. In one study, no haemodialysis patient in the Northern Territory of Australia receiving daily prophylactic trimethoprim/sulfamethoxazole (TMP-SMX) 160/800mg in the local wet season developed melioidosis, compared to a rate of 17.4% in patients not receiving this therapy; side effects of the TMP-SMX in this cohort were uncommon and mild [12]. The success of this strategy and clinical experience in the Northern Territory has led to Australian national guidelines also recommending TMP-SMX to prevent melioidosis in patients with significant immunosuppression [13]. Daily TMP-SMX (at a dose of 160/800mg) is recommended in individuals with significant immunosuppression (defined as prednisone 20mg daily or equivalent or other potent immunosuppressive drug therapy such as chemotherapy) if they have lived in tropical Australia and have positive serology for *B. pseudomallei* or if they have a history of melioidosis. The guidelines also recommend consideration of daily wet season TMP-SMX prophylaxis in individuals with significant immunosuppression and negative *B. pseudomallei* serology who live in, or visit, an area where melioidosis is endemic.

As a result, in the Top End of the Northern Territory, where the incidence of melioidosis is as high as 51.2/100,000 population, patients with cancer receiving immunosuppressive anti-cancer therapy routinely receive TMP-SMX prophylaxis [14]. However, differences in demographics, the burden of comorbidities, and the patterns of environmental exposure means that the incidence of melioidosis is lower in other tropical Australian regions [14–16]. The lower incidence of melioidosis and the potential for severe, and even life-threatening, complications from TMP-SMX prophylaxis, means that the risks and benefits of TMP-SMX prophylaxis for melioidosis in different regions need to be balanced carefully [17].

The incidence of melioidosis in Far North Queensland (FNQ), in tropical Australia, has risen fourfold in the last 25 years; in 2022 the incidence reached 20.4/100,000 population [18]. The region also has a significant incidence of new cancer diagnoses which reached 690.9/100,000 in 2021 [19]. However, it has not been local practice to prescribe TMP-SMX prophylaxis for melioidosis to patients with cancer receiving anti-cancer therapy in FNQ. Local clinicians have not identified a high risk of melioidosis in patients receiving anti-cancer therapy and have concerns about the possible side effects of TMP-SMX prophylaxis–particularly myelosuppression–as well as the potential for medication interactions and polypharmacy. However, with the recent rise in the local incidence of melioidosis the benefits of TMP-SMX prophylaxis may now outweigh these putative risks.

This study had two separate aims. The first was to describe the burden, clinical characteristics, comorbidities, and outcomes of melioidosis in patients with cancer to define the clinical phenotype of melioidosis in this population more precisely. The second aim was to identify if TMP-SMX chemoprophylaxis was associated with a lower rate of subsequent melioidosis in patients receiving anti-cancer therapies. This might help determine if there may be a role for chemoprophylaxis against melioidosis in patients receiving anti-cancer therapies in the FNQ region.

## Methods

### Ethics statement

The Far North Queensland Human Research Ethics Committee provided ethical approval for the study (HREC/15/QCH/46–977). As the data were deidentified the committee waived the requirement for informed consent.

### Study setting

FNQ in tropical Australia spans an area of 380,000 km$^2$ and a population of 290,000, 17% of whom identify as Aboriginal and/or Torres Strait Islander Australians (hereafter respectfully described as First Nations Australians). Most of the FNQ population live in the administrative hub of Cairns, which has the region's major hospital. Cairns Hospital has tertiary level oncology and haematology services and is the referral centre for the management of all solid organ and haematological malignancies in the FNQ region. The Cairns Hospital's departments of oncology and haematology prescribe all chemotherapy in the region and provide support to satellite regional hospitals where some lower intensity chemotherapy is administered. There are only two health services in FNQ, the Cairns and Hinterland Hospital and Health Service (CHHHS)–which serves Cairns and the surrounding region–and the Torres and Cape Hospital and Health Service (TCHHS) which serves the population living on the Cape York Peninsula and the Torres Strait Islands.

### Characteristics of patients with cancer and melioidosis

To define the clinical characteristics, comorbidities, and outcomes of melioidosis in patients with cancer we identified every patient with *B. pseudomallei* cultured in the Cairns Hospital laboratory between the 1st of January 1998 and 1st June 2023. Melioidosis is a notifiable disease in the state of Queensland and the Cairns Hospital laboratory confirms the diagnosis of all cases of melioidosis in the FNQ region. Data were collected and entered prospectively into a dedicated database from October 1st, 2016, while patients presenting prior to this date had their data entered into the database retrospectively. The patients' medical records were reviewed, and their demographics and medical history were recorded. All individuals receiving care in Queensland's public health system are asked whether they identify as a First Nations Australian. Patients living in the TCHHS were said to have a remote residence. The presence of comorbidities that predispose to melioidosis were specifically sought as described previously [20]. Melioidosis was categorised as having an acute (symptoms present for <2 months) or chronic (symptoms present ≥2 months) presentation [14]; the clinical phenotype of their melioidosis was also determined (S1 Table).

The medical records were also examined thoroughly for any diagnosis of concurrent active cancer. This included the type of malignancy (either solid organ or haematological) and the presence of metastatic or non-localised disease. The chart was reviewed to identify if the melioidosis was identified before or after the cancer diagnosis. If the cancer was diagnosed

before the melioidosis it was determined if the patient had received anti-cancer therapy in the 12 months prior to their melioidosis diagnosis.

The treatment and clinical course of the individuals' melioidosis was recorded. In the event of death, it was determined whether this was more likely due to melioidosis or the underlying cancer. Relapse was defined as recurrence of melioidosis after completion of the eradication phase of antibiotic therapy.

### Utility of TMP-SMX chemoprophylaxis in patients receiving anti-cancer therapy

In order to define the potential utility of TMP-SMX chemoprophylaxis for melioidosis in patients with cancer receiving treatment for their cancer, we interrogated the oncology software programme MOSAIQ used at Cairns Hospital to identify all individuals receiving anti-cancer therapy in the region; data were only available after 2008, when use of the MOSAIQ software commenced at the hospital. Systemic myelosuppressive anti-cancer therapies were defined as of mild, moderate, or high intensity using National Comprehensive Cancer Network guidelines [21]. Mild intensity therapy included single agent treatments with low rates of myelosuppression. Moderate intensity therapy included single or multi-agent regimens where myelosuppression is a possible, but not common, complication. High intensity therapy was defined as a regimen that was likely to cause significant or prolonged myelosuppression. The patients who received non-suppressive anti-cancer treatments–including immunotherapy and targeted therapies–were also identified. Many patients received multiple lines of anti-cancer therapy during the study period; the highest intensity regimen that the patient received was recorded in these cases.

The number of patients receiving treatment for cancer who also received TMP-SMX pro-phylaxis–which in this cohort was prescribed at a dose of 160/800mg twice a day on two days a week for protocol-related prevention of *Pneumocystis jirovecii* pneumonia–was determined from pharmacy records. We then cross-referenced these data with the individuals who were diagnosed with melioidosis during the same time period to identify if there was any association between the prescription of TMP-SMX prophylaxis for *P. jirovecii* pneumonia and a reduced subsequent incidence of culture-confirmed melioidosis.

### Statistical analysis

Data were de-identified, entered into an electronic database (Microsoft Excel) and stored in accordance with Australian guidelines for the responsible conduct of research [22]. These data were analysed using statistical software (Stata version 14.2). Univariate analysis was performed using Fisher's exact test or logistic regression. Multivariate analysis of the characteristics of the patients with melioidosis who did–and did not–have cancer was performed using logistic regression and a backwards stepwise approach; variables were selected for consideration in the multivariate model if their p value in univariate analysis was <0.10. If individuals were missing data, they were not included in the analysis examining those variables. Queensland's public health system uses Australian Bureau of Statistics census data to calculate estimated resident populations of the CHHHS and the TCHHS; these estimates were used to determine disease incidence [23]. Trends over time in the incidence of melioidosis and in the proportion of individuals with melioidosis an active cancer were analysed using Spearman's test for correlation.

### Results

There was a total of 477 cases melioidosis during the study period. The incidence increased from 4.6/100000 in 1998 to 20.4/100000 in 2022 –the first and last complete calendar year in the study period (p<0.001, $r_s$ = 0.80). The incidence in the CHHHS increased from 1.5/100000

in 1998 to 20.3/100000 in 2022 (p<0.001, $r_s$ = 0.65), but it remained stable in the TCHHS (mean incidence (95% confidence interval): 24.7 (20.3–29.1)/100,000) during the entire study period. In 2022, the incidence in the local First Nations Australians population was 38.8/100,000 compared with 16.6/100,000 in the local non-Indigenous population.

In 446/477 (94%) it was possible to determine if there was an active malignancy at the time of diagnosis of melioidosis; an active cancer was confirmed in 47/446 (11%). The proportion of individuals with melioidosis and an active cancer increased during the study period (p = 0.0499, $r_s$ = 0.09); in the first five years of the study period 0/28 had an active cancer, compared to 29/234 (12%) in the last 5 years.

The characteristics of the patients with melioidosis who did–and did not–have cancer are compared in Table 1. In multivariate analysis, patients with melioidosis and cancer were more likely to be older (odds ratio (OR) (95% confidence interval (CI)): 1.05 (1.03–1.08), p<0.0001) and immunosuppressed (OR (95% CI): 11.54 (5.41–24.6), p<0.0001), but less likely to have diabetes mellitus (OR (95% CI): 0.44 (0.20–0.99), p = 0.048).

Of the 47 individuals with an active cancer, 37 (79%) had a solid organ tumour and 11 (23%) had a haematological malignancy (1 patient had a solid organ tumour and haematological malignancy concurrently). Individuals with a solid organ tumour were more likely to identify as a First Nations Australian and smoke cigarettes at the time of diagnosis of their melioidosis, but these differences failed to reach statistical significance (p = 0.09 and p = 0.08 respectively using Fisher's exact test) (S2 Table).

## Patients with a solid organ tumour

In 30/37 (81%) with a solid organ tumour, the cancer was already diagnosed and in 7 it was diagnosed during the evaluation and management of the melioidosis. There were 33/35 (94%) who presented with acute symptoms, 2/35 (6%) had chronic symptoms; in 2/37 (5%) the duration of symptoms could not be determined. No cases were thought to have activated from latency.

Most (20/37, 54%) patients with solid organ tumours had metastatic disease. The most common diagnoses were prostate cancer (12/37, 32%) and lung cancer (9/37, 24%). Myelosuppressive anti-cancer therapy had been prescribed to 11/37 (31%) in the 12 months prior to their diagnosis of melioidosis. There were 3/37 (8%) who had received radiotherapy in the 12 months prior their diagnosis of melioidosis, 2 of whom had not received chemotherapy, but both these patients had other risk factors for melioidosis. Indeed, there were only 8/37 (21%) with no other risk factors for melioidosis who had not received myelosuppressive anti-cancer therapy in the prior 12 months (Table 2).

Twelve months after the diagnosis of melioidosis, 19/37 (51%) were still alive; 7/18 (39%) deaths were due to melioidosis and 11/18 (61%) were due to the underlying cancer. The 29 patients who completed their intensive therapy for melioidosis received a median (interquartile range (IQR)) of 4 (2–4) weeks of intravenous therapy. Most (22/29, 76%) subsequently received 3 months of eradication therapy with TMP-SMX; four patients had a longer course, 1 patient had no eradication (due to a decision to transition to palliative care) and in two the duration of eradication therapy was uncertain. Skin rash necessitated a switch from TMP-SMX to doxycycline in 4/29 (14%); 1 of these 4 patients represented with melioidosis 4 years later, although it was not possible to perform multi-locus sequence typing to determine if this was relapse or reinfection.

## Haematology cohort

There were 11 individuals with a haematological malignancy; the malignancy was diagnosed already in 10 and it was diagnosed in the other during the evaluation of the melioidosis. Patients presented acutely in 8 cases while 3 patients presented with chronic symptoms; no

**Table 1. Characteristics of patients with–and without an active cancer–in the cohort.**

| | All n = 446 [a] | Active cancer n = 47 | No active cancer n = 399 | Odds ratio (95% confidence interval) | P [b] |
|---|---|---|---|---|---|
| Age (years) | 55 (43–66) | 67 (58–73) | 53 (42–65) | 1.05 (1.03–1.07) | <0.0001 [d] |
| Child (<16 years) | 21 (5%) | 0 | 21 (5%) | - | - |
| Male sex | 313 (70%) | 35 (74%) | 278 (70%) | 1.27 (0.64–2.53) | 0.50 |
| First Nations Australians | 207 (46%) | 9 (19%) | 198 (50%) | 0.24 (0.11–0.51) | <0.0001 [d] |
| Remote residence [c] | 150 (33%) | 8 (17%) | 142 (36%) | 0.37 (0.17–0.82) | 0.01 |
| Wet season presentation | 332 (74%) | 34 (72%) | 298 (75%) | 0.89 (0.45–1.75) | 0.73 |
| Diabetes mellitus | 226 (51%) | 11 (23%) | 215 (54%) | 0.26 (0.13–0.53) | <0.0001 [d] |
| Hazardous alcohol use | 170/440 (39%) | 10/46 (22%) | 160/394 (41%) | 0.41 (0.20–0.84) | 0.02 [d] |
| Smoker | 220/440 (50%) | 20 (43%) | 200 (51%) | 0.75 (0.40–1.38) | 0.35 |
| Chronic lung disease | 95 (21%) | 17 (36%) | 78 (20%) | 2.33 (1.22–4.44) | 0.01 [d] |
| Chronic kidney disease | 55 (12%) | 4 (9%) | 51 (13%) | 0.63 (0.22–1.84) | 0.40 |
| Immunosuppression | 58/327 (18%) | 28 (60%) | 30/280 (11%) | 12.20 (6.13–24.60) | <0.0001 [d] |
| Lung involvement | 325/444 (73%) | 35 (74%) | 290 (73%) | 1.08 (0.54–2.15) | 0.84 |
| Genitourinary involvement | 88/439 (20%) | 6 (13%) | 82 (21%) | 0.55 (0.23–1.35) | 0.19 |
| Musculoskeletal involvement | 55/438 (13%) | 3 (6%) | 52 (13%) | 0.44 (0.13–1.48) | 0.19 |
| SSTI | 63/438 (14%) | 2 (4%) | 61/391 (16%) | 0.24 (0.06–1.02) | 0.053 [d] |
| CNS involvement | 16/438 (4%) | 0 | 16/391 (4%) | - | - |
| Bacteraemia | 314 (70%) | 38 (81%) | 276 (69%) | 1.88 (0.88–4.01) | 0.10 |
| Septic shock | 91/428 (21%) | 8/46 (17%) | 83/382 (22%) | 0.76 (0.34–1.69) | 0.57 |
| ICU admission | 107 (24%) | 7 (15%) | 100 (25%) | 0.52 (0.23–1.21) | 0.13 |
| Died from melioidosis before hospital discharge | 40 (9%) | 7 (15%) | 33 (8%) | 1.94 (0.81–4.67) | 0.14 |
| Relapse of melioidosis | 10 (4%) | 1 (2%) | 9 (2%) | 0.94 (0.12–7.60) | 0.96 |

SSTI: Skin and soft tissue infection. CNS: Central nervous system. ICU: Intensive Care Unit.

[a] It was possible to reliably confirm or exclude an active cancer in only 446/477 patients in the cohort. Retrospective data collection from cases before October 2016 resulted in some missing data prior to this time and, accordingly, a difference in the denominator for some variables.

[b] Univariate analysis

[c] Patients living in the Torres and Cape Hospital and Health Service

[d] Included in the multivariate analysis.

cases were thought to have activated from latency. The most common diagnosis was a lympho-proliferative disorder (5/11, 45%). Most (9/11, 82%) had non-localised disease and 5/11 (45%) were receiving anti-cancer therapy at the time melioidosis was diagnosed. This was myelosup-pressive chemotherapy in 3/11 (27%), high dose corticosteroids in 1/11 (9%) and a Janus kinase inhibitor in 1/11 (9%). There were 2/11 (18%) with no other risk factors for melioidosis who had not received myelosuppressive anti-cancer therapy in the prior 12 months (Table 3).

The patients received a median (IQR) of 3 (4–6) weeks of intensive intravenous therapy and either 3 or 6 months of eradication therapy. Second-line therapy was used in 3 cases due to TMP-SMX related side effects. At 12 months 7/11 (64%) were still alive, 2 deaths were attributable to melioidosis.

## Association between TMP-SMX prophylaxis during anti-cancer therapy and the subsequent diagnosis of melioidosis

Between 1 January 2008 and June 1, 2023, there were 4400 patients who received myelosup-pressive anti-cancer therapy and 368 individuals who received non-immunosuppressive anti-

**Table 2. Characteristics and outcomes for patients with melioidosis and a solid organ cancer.**

| Age sex | Cancer type | Metastatic | Other risk factors for melioidosis | Clinical phenotype | Bacteraemia | Status at 12 months | Anti-cancer therapy in prior 12 months |
|---|---|---|---|---|---|---|---|
| 44M | Lung [a] | No | Alcohol, CLD | Pneumonia | Yes | Alive | No |
| 50M | Lung | Yes | No | Pneumonia | Yes | Died from malignancy | Carboplatin, etoposide |
| 62M | Lung | Yes | DM, CLD, CKD | Pneumonia | Yes | Died from malignancy | No |
| 65M | Lung | Yes | No | Pneumonia | Yes | Died from melioidosis | Carboplatin |
| 66M | Lung | No | CLD | Pneumonia, prostate | Yes | Alive | Sotorasib |
| 67F | Lung [a] | Yes | CLD | Pneumonia | Yes | Died from malignancy | Carboplatin, paclitaxel, pembrolizumab |
| 69M | Lung | Yes | CLD | Pneumonia | Yes | Alive | Carboplatin, etoposide |
| 71M | Lung, supraglottic SCC | No | No | Pneumonia, prostate | No | Died from malignancy | No |
| 84M | Lung | No | Alcohol, CLD | Pneumonia | No | Died from melioidosis | No |
| 67M | Prostate | Yes | CLD | Pneumonia | Yes | Alive | No |
| 70M | Prostate [c] | Yes | Alcohol, CLD | Pneumonia | Yes | Alive | ADT |
| 71M | HCC, prostate | No | Alcohol, CLD | Prostate | Yes | Died from malignancy | No |
| 72M | Prostate | Yes | No | Pneumonia | No | Alive | No |
| 73M | Prostate and rectal [c] | No | DM, alcohol | Pneumonia | No | Alive | Capecitabine and ADT |
| 75M | Prostate | Yes | CLD | Pneumonia, renal | No | Died from melioidosis | No |
| 78M | Prostate and NHL | Yes | CLD, DM | Pneumonia | Yes | Alive | No |
| 78M | Prostate and bladder | No | No | Pneumonia | Yes | Alive | No |
| 80M | Prostate [a] | Yes | CLD | Pneumonia | Yes | Alive | No |
| 81M | Prostate | No | DM | Pneumonia, SSTI | Yes | Died from melioidosis | ADT |
| 83M | Prostate | Yes | DM | Bacteraemia no focus | Yes | Died from malignancy | ADT |
| 78M | Prostate | Yes | Alcohol | Bacteraemia | Yes | Alive | No |
| 46M | Renal | Yes | CLD | Disseminated | Yes | Died from malignancy | Cabozantinib |
| 56M | Renal | No | No | Bacteraemia no focus | Yes | Alive | No |
| 42F | Breast | Yes | No | Bacteraemia no focus | Yes | Alive | Doxorubicin, cyclophosphamide |
| 48F | Breast | No | No | Bacteraemia | Yes | Alive | Doxorubicin, cyclophosphamide |
| 47M | HCC [a] | No | DM, alcohol, CKD | Pneumonia | Yes | Died from melioidosis [b] | No |
| 68M | HCC | Yes | No | Pneumonia | Yes | Died from melioidosis | No |
| 58M | Colorectal [a] | Yes | DM | Pneumonia, liver | Yes | Alive | Capecitabine, FOLFIRI |
| 79F | Colorectal | No | No | SSTI | No | Alive | No |
| 65M | Oesophageal | No | No | Pneumonia | No | Died from malignancy | No |
| 47M | Gastric | Yes | No | Pneumonia | Yes | Died from melioidosis | Irinotecan capecitabine, FOLFOX, trastuzumab |
| 59F | Cholangiocarcinoma [a] | No | DM, CKD | Bacteraemia no focus | Yes | Died from malignancy | No |
| 51F | Cervical [a] | Yes | No | Pneumonia | Yes | Died from malignancy | No |
| 68F | Endometrial | No | No | Pneumonia | Yes | Alive | Carboplatin, paclitaxel |
| 64F | Ovarian | Yes | No | Pneumonia, adnexal collection | Yes | Died from malignancy | Gemcitabine |
| 65F | Thymoma [c] | No | DM | Pneumonia | Yes | Alive | No |
| 61M | Tonsillar | No | Alcohol, CLD | Pneumonia, prostate | Yes | Alive | No |

M: Male; F: Female; NHL: Non-Hodgkins lymphoma; SCC: squamous cell carcinoma; HCC: hepatocellular carcinoma; DM: diabetes mellitus; CKD: chronic kidney disease; CLD: chronic lung disease; SSTI: skin and soft tissue infection; ADT: Androgen deprivation therapy; FOLFIRI: fluorouracil, leucovorin, irinotecan; FOLFOX: fluorouracil, leucovorin, oxaliplatin.

[a] Cancer was unknown at the time of the diagnosis of the melioidosis and was only diagnosed during inpatient evaluation.

[b] Death occurred due to a presumed relapse of melioidosis (clinical diagnosis, not culture proven) 4 months after their initial diagnosis of melioidosis.

[c] Received radiotherapy in the 12 months prior to diagnosis of culture proven melioidosis

**Table 3. Characteristics and outcomes for patients with melioidosis and a haematological malignancy.**

| Age, sex | Which malignancy | Localised disease | Other risk factors for melioidosis | Clinical phenotype | Bacteraemia | Status at 12 months | Anti-cancer therapy in prior 12 months |
|---|---|---|---|---|---|---|---|
| 59F | Mantle cell lymphoma | No | CLD | Pneumonia | Yes | Died from melioidosis | HyperCVAD, venetoclax |
| 64M | Diffuse large B-cell lymphoma with previous autologous stem cell transplant | No | No | Pneumonia | Yes | Died from malignancy | Rituximab, gemcitabine, oxaliplatin, high dose corticosteroids |
| 71F | B-cell lymphoproliferative disorder | No | Alcohol | Disseminated | Yes | Alive | High dose corticosteroids |
| 78M | Non-Hodgkins lymphoma and prostate cancer | No | DM, CLD | Pneumonia | Yes | Alive | No |
| 78M | Large B-cell lymphoma [a] | No | DM | Bacteraemia—no focus | Yes | Alive | No |
| 58M | Acute myeloid leukaemia | Yes | No | Pneumonia | Yes | Alive | No |
| 71M | Chronic lymphatic leukaemia | Yes | Alcohol | SSTI | No | Alive | No |
| 50F | Myelofibrosis | No | No | Bacteraemia | Yes | Alive | Ruxolitinib |
| 56M | Myelofibrosis | No | No | Bacteraemia—no focus | Yes | Alive | No |
| 67M | Myelodysplastic syndrome | No | No | Pneumonia | Yes | Died from malignancy | Rituximab, azacitidine, low dose corticosteroids |
| 84M | Multiple myeloma | No | CLD, DM | Pneumonia | No | Died from melioidosis [a] | No |

M: Male; F: Female; DM: diabetes mellitus; CKD: chronic kidney disease; CLD: chronic lung disease; SSTI: skin and soft tissue infection; AML: acute myeloid leukaemia. HyperCVAD: **C**yclophosphamide, **v**incristine, doxorubicin, **d**examethasone alternating with methotrexate and cytarabine. High dose corticosteroids: >20mg prednisolone or equivalent

[a] Cancer was unknown at the time of the diagnosis of the melioidosis and was only diagnosed during inpatient evaluation.

cancer therapy. The incidence of melioidosis in patients receiving this anti-cancer therapy, stratified by myelosuppressive intensity of the regimen, is presented in Table 4. There was no statistical difference in the proportion of individuals who subsequently developed melioidosis among those who did–and did not–receive TMP-SMX prophylaxis for *P. jirovecii* pneumonia during their anti-cancer therapy. The characteristics of the patients who did–and did not–have immune suppressing anti-cancer therapy prior to the diagnosis of melioidosis (stratified by whether they did–or did not–receive TMP/SMX prophylaxis) is presented in S3 Table.

## Discussion

Despite a significant and rising local incidence of melioidosis, cancer was not a common pre-disposing factor for the infection in this region of tropical Australia. Indeed, cancer was the sole risk factor for melioidosis in only 2.2% of patients with culture-confirmed melioidosis during the 25-year period, a figure that rose to 4.5% if patients receiving immunosuppressing anti-cancer therapy in the prior 12 months were also considered. Although FNQ clinicians did not routinely prescribe TMP-SMX prophylaxis to local patients receiving anti-cancer thera-pies, melioidosis was a rare complication in this population. Indeed, there was no statistical difference in the subsequent incidence of melioidosis between patients who did–and did not–receive protocol based TMP-SMX prophylaxis for *P. jirovecii* pneumonia during the study period. While side effects from TMP-SMX prophylaxis are rare, they may occasionally be life-threatening [24]. Our data therefore do not presently support the prescription of TMP-SMX prophylaxis for melioidosis in patients receiving anti-cancer therapies in this region of Australia.

**Table 4. Proportion of patients receiving TMP-SMX prophylaxis for _P. jirovecii_ pneumonia as part of their anti-cancer therapy and the subsequent incidence of melioidosis (stratified by myelosuppressive intensity of the regimen).**

| | High intensity chemotherapy regimen | Medium intensity chemotherapy regimen | Low intensity chemotherapy regimen | Overall myelosuppressive anti-cancer therapy [a] | Non myelosuppressive anti-cancer therapy [b] | Overall anti-cancer therapy [c] |
|---|---|---|---|---|---|---|
| Total number of patients receiving anti-cancer therapy | 385 | 2947 | 1068 | 4400 | 368 | 4768 |
| Patients who received TMP-SMX prophylaxis [d] | 227 (59%) | 482 (16%) | 28 (3%) | 737 (17%) | 0 | 737 (15%) |
| Patients who did not receive TMP-SMX prophylaxis [d] | 158 (41%) | 2465 (84%) | 1040 (97%) | 3663 (83%) | 368 | 4031 (85%) |
| Number of melioidosis cases in patients receiving prophylaxis | 1/227 (0.44%) | 0/482 (0%) | 0/28 (0%) | 1/737 (0.15%) | - | 1/737 (0.14%) |
| Number of melioidosis cases in patients not receiving prophylaxis | 2/158 (1.27%) | 8/2465 (0.32%) | 6/1040 (0.58%) | 16/3663 (0.44%) | 0/368 (0%) | 16/4031 (0.40%) |
| Risk ratio (95% confidence interval) | 0.35 (0.03–3.81) | - | - | 0.31 (0.04–2.34) | - | 0.34 (0.05–2.57) |
| p value [e] | 0.37 | 0.24 | 0.85 | 0.20 | - | 0.24 |
| Number of fatal melioidosis cases in patients receiving TMP-SMX prophylaxis | 0/227 (0%) | 0/482 (0%) | 0/28 (0%) | 0/737 (0%) | - | 0/737 (0%) |
| Number of fatal melioidosis cases in patients not receiving TMP-SMX prophylaxis | 1/158 (0.63%) | 1/2465 (0.04%) | 1/1040 (0.10%) | 3/3663 (0.08%) | 0/368 (0%) | 3/4031 (0.07%) |
| p value [e] | 0.41 | 0.84 | 0.97 | 0.58 | - | 0.60 |

[a] Including only low, medium, and high intensity myelosuppressive anti-cancer therapy.

[b] Receiving only non-myelosuppressive anti-cancer therapy (e.g. immunotherapy, non-myelosuppressive targeted therapy)

[c] All patients receiving anti-cancer therapy.

[d] TMP-SMX prophylaxis for _Pneumocystis jirovecii_ pneumonia as per the anti-cancer therapy protocol (160/800mg twice a day on Monday and Thursday)

[e] Calculated using one-sided Fisher's exact test

The proportion of cases in which cancer contributed to the development of melioidosis increased slightly in our cohort during the study period and can be explained by recent changes in the local incidence of the disease. Recent urban expansion is hypothesised to have contributed to a 10-fold increase in the incidence of melioidosis in the city of Cairns, a population where the people are older–and, accordingly, at greater risk of cancer–than the population living in remote FNQ [15,19,25]. Although patients with melioidosis and cancer were more likely to be immunosuppressed, this was often a result of the anti-cancer therapy that they were receiving for their malignancy. The fact that diabetes mellitus was less common among melioidosis patients with an active cancer is likely to be explained by the significant burden of diabetes in the younger First Nations population in the region who are more likely to have diabetes and to live remotely [26,27]. A lower rate of skin and soft tissue disease among patients with cancer is likely to be explained by the fact that SSTI occurs more commonly in patients without any underlying risk factors [28]. The association with chronic lung disease is likely to be explained by the critical role that cigarette smoking plays in the development of both cancer and chronic lung disease. Although patients with melioidosis and cancer were commonly not currently smoking or drinking alcohol hazardously, this may reflect changes in lifestyle precipitated by their diagnosis and treatment [29].

The cancer in the patients with melioidosis and cancer in this cohort was usually already diagnosed and the proportion of different cancers in the patients with melioidosis was similar to that in the local general population [19]. There was no apparent relationship between the

patients' cancer diagnosis and the clinical phenotype of their melioidosis. Although all the patients with lung cancer had melioidosis involving the lung, melioidosis involved the lungs in almost three-quarters of the entire cohort. Furthermore, most patients with lung cancer also had underlying chronic lung disease, itself a risk factor for pulmonary melioidosis [14].

Over 90% of the patients with an active cancer presented acutely with symptoms of melioidosis. Although it is usually impossible to determine if a patient's presentation represents reactivation of latent infection, the fact that there was no difference in wet season presentation between patients with and without an underlying malignancy suggests that reactivation of latent infection was not common in our cohort.

Death from melioidosis before hospital discharge was more common in patients with melioidosis and cancer than in those without cancer, although this difference failed to reach statistical significance. Relapse of melioidosis was uncommon in patients completing their eradication treatment–occurring in only one patient who received the second-line eradication therapy doxycycline–suggesting that long term secondary prophylaxis may not be necessary in most patients who are able to complete standard eradication therapy. At 12 months, patients were more likely to have died from their underlying cancer (which was often metastatic) than their melioidosis, emphasising the significant contribution that comorbidity makes to the long term outcomes of people who survive their melioidosis [30].

There was no evidence in our study to support routine TMP-SMX use for primary prophylaxis of melioidosis in cancer patients receiving anti-cancer therapy in this region of Australia. This echoes the findings of a study that examined local rheumatological patients taking immunomodulatory therapy [31]. However, prophylactic TMP-SMX has potential benefits beyond reducing the incidence of melioidosis. It is prescribed primarily for prevention of *P. jirovecii* pneumonia–the reason for its prescription in this cohort–and may have salutary effects against other local opportunistic infections including Nocardia and Listeria [32,33]. Prophylactic TMP-SMX reduces the incidence of bacterial sepsis in patients with transplants and there is also growing interest in the utility of prophylactic TMP-SMX for other common infections in tropical Australia including group A Streptococcus and *Staphylococcus aureus* [34–36].

However, these putative benefits need to be balanced against the potential for harm from expanded prescription of TMP-SMX. The reported incidence of adverse reactions to TMP-SMX chemoprophylaxis varies significantly in different populations Only 1.3% of an Australian haemodialysis population ceased TMP-SMX due to side effects [12], although an incidence of adverse reactions to TMP-SMX of 41.5/100-patient years and discontinuation rates of 14.5% and have been reported in the Asian rheumatology population [37,38]. While almost all these adverse reactions are mild and less frequent with lower dosing, life-threatening reactions such as Stevens-Johnson Syndrome and toxic epidermal necrolysis are well recognised [39,40]. Pancytopenia may also be more common in patients receiving myelosuppressive chemotherapy, which can be potentially fatal or can, at the very least, interrupt ongoing anti-cancer therapy [37].

Expanded use of prophylactic TMP-SMX has the potential to increase resistance to an agent which has an important role in tropical Australia [41–43]. *S. aureus* is the commonest cause of sepsis requiring ICU admission in the FNQ region and up to a third of these isolates are now methicillin-resistant with 10% of blood stream isolates resistant to TMP-SMX [27,44,45]. Widespread use of prophylactic TMP-SMX could increase the proportion of resistant isolates further [46]. Evolving antimicrobial resistance–which is driven largely by over-prescription of antibiotics–is an important threat for the continuing success of cancer therapy [47,48]. Infection plays a primary or associated role in the cause of death in approximately 50% of cancer patients [49], and in one meta-analysis the identified pathogen in over a quarter

of the infections complicating anti-cancer therapy, was resistant to the standard prophylactic antibiotics that had been prescribed [50].

Concomitant prescription of antibiotics and immune checkpoint inhibitors is associated with significantly worse treatment-related outcomes [51]. This has important implications in a region where immune checkpoint inhibitors are commonly prescribed to patients with lung cancer and melanoma two of the five most common cancers in the region [19]. Finally, there have been significant recent global shortages of some oral antibiotics including TMP-SMX. Expanded use of TMP-SMX prophylaxis has the potential to threaten the supply of the antibiotic for the treatment of important pathogens in tropical Australia and is an additional concern [52].

The study has several significant limitations. The study's retrospective nature precluded comprehensive data collection on the melioidosis cases prior to 2016 and anti-cancer therapy regimens prior to 2008. The study period was a long one and approaches to the treatment of both melioidosis and malignancy have evolved over this time [14,53,54]. The patients with a malignancy were a highly heterogeneous population–there is a significant difference between a patient with diffuse large B-cell lymphoma and a previous stem cell transplant receiving multi-agent chemotherapy and a patient with localised prostate cancer who is managed with watchful waiting. Melioidosis serology–which is recommended in Australian guidelines to inform melioidosis risk–was not collected, although the limitations of serology in areas that are endemic for melioidosis are well known [55]. Other factors that increase the risk of melioidosis in the patients receiving anti-cancer therapies (comorbidities, environmental exposures) were not collected, however, our findings support the clinical observation that melioidosis is a rare complication of anti-cancer therapy in FNQ. Indeed, many patients received several lines of chemotherapy, so the risk of melioidosis per course of chemotherapy would be even lower than we have reported. Although our data therefore do not presently support the routine prescription of TMP-SMX prophylaxis for melioidosis in cancer patients receiving anti-cancer therapy in this region of Australia, it may have utility in regions where the incidence of melioidosis is higher–such as the Top End of the Northern Territory–and in patients who are also at very high risk of other opportunistic infections. It may also have utility in the prevention of other life-threatening infections, data that we were unable to collect systematically. As ever, it will be a decision for local clinicians to consider the benefits and risks of such an approach [12,17]. If the local incidence of melioidosis continues to rise, TMP-SMX prophylaxis may become more appealing, although short term reductions in the incidence of infection will always need to be balanced against the risk of driving antimicrobial resistance in this high-risk population [47,48]. It was also notable that there was only 1 case of melioidosis–and no cases of fatal melioidosis–among individuals taking a dose of 160/800mg twice a day on two days in a week (a lower dose than the 160/800 daily recommended in current Australian guidelines) [13]. It may be that this lower dose may also be effective for the prevention of melioidosis in jurisdictions that choose to prescribe TMP-SMX chemoprophylaxis for melioidosis.

## Conclusions

We have presented the characteristics and clinical course of 47 cases of culture proven melioidosis in patients with active malignancy in tropical Australia. In Australia's well-resourced health system less than 15% of these individuals died from melioidosis before discharge and disease relapse was uncommon, however, almost a third of the patients surviving their melioidosis had died from their malignancy within 12 months of their infection's diagnosis. Local clinicians need to consider the risks and benefits of TMP-SMX prophylaxis for melioidosis in cancer patients receiving anti-cancer therapies, but our data do not presently support its routine prescription in this region of Australia.

## Supporting information

**S1 Table. Definitions of the clinical phenotypes of melioidosis used in the study.**
(DOCX)

**S2 Table. Comparison of the characteristics of the individuals with a solid organ tumour and those with a haematological malignancy.**
(DOCX)

**S3 Table. Comparison of the characteristics of the patients who had no cancer therapy prior to the diagnosis of melioidosis and the patients that did (stratified by whether they did–or did not–take TMP/SMX prophylaxis).**
(DOCX)

## Acknowledgments

The authors would like to acknowledge the work of the many health care workers involved in the care of the patients.

## Author Contributions

**Conceptualization:** Simon Smith, Patrick Donald, Josh Hanson.

**Data curation:** Tej Shukla, Simon Smith, Kristoffer Johnstone.

**Formal analysis:** Josh Hanson.

**Investigation:** Tej Shukla, Kristoffer Johnstone, Josh Hanson.

**Methodology:** Josh Hanson.

**Supervision:** Simon Smith, Patrick Donald.

**Visualization:** Josh Hanson.

**Writing – original draft:** Tej Shukla, Josh Hanson.

**Writing – review & editing:** Simon Smith, Kristoffer Johnstone, Patrick Donald, Josh Hanson.

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
