## [Decision Letter · Decision Letter 0]

16 Aug 2024

Dear Dr. Hanson,

Thank you very much for submitting your manuscript "The characteristics and clinical course of patients with melioidosis and cancer" for consideration at PLOS Neglected Tropical Diseases. As with all papers reviewed by the journal, your manuscript was reviewed by members of the editorial board and by several independent reviewers. In light of the reviews (below this email), we would like to invite the resubmission of a significantly-revised version that takes into account the reviewers' comments. 

Reviewer 3 raises multiple concerns relating to methodology, analysis, ethics, and presentation/interpretation. Please carefully address each of these concerns in your revision.

We cannot make any decision about publication until we have seen the revised manuscript and your response to the reviewers' comments. Your revised manuscript is also likely to be sent to reviewers for further evaluation.

Sincerely,

T Eoin West, MD, MPH

Guest Editor

Ana LTO Nascimento

Section Editor

Reviewer 3 raises multiple concerns relating to methodology, analysis, ethics, and presentation/interpretation. Please carefully address each of these concerns in your revision.

Reviewer's Responses to Questions

**Key Review Criteria Required for Acceptance?**

**Methods**

-Are the objectives of the study clearly articulated with a clear testable hypothesis stated?

-Is the study design appropriate to address the stated objectives?

-Is the population clearly described and appropriate for the hypothesis being tested?

-Is the sample size sufficient to ensure adequate power to address the hypothesis being tested?

-Were correct statistical analysis used to support conclusions?

-Are there concerns about ethical or regulatory requirements being met?

Reviewer #1: Yes

Reviewer #2: Yes, method and statistical analysis used are appropriate.

Reviewer #3: 1. No hypothesis is provided for the study (usually in the introduction section but is a component of study methods).

2. The study design is confusing as they state that it is prospective and retrospective. It appears as though hospital databases are used for the study which then makes this fully retrospective. If it was prospective, I do not see that patients were approached for consent as they would be individual admissions. If the data was collected for example after someone's admission was done, that still makes it retrospective.

3. The databases and variables collected for the study should be described in one paragraph/section to make it clear what data was obtained from where/how and any definitions that may be relevant for socio-demographic and clinical variables. I think the clinical diagnosis and definitions of melioidosis does need to be described here like phenotypes/syndromes as it is relevant to the study itself and at minimum be included in a supplement.

4. Where was the data compiled prior to de-identification? Were the identifying variables stored anywhere as a master dataset for linking? 

5. Greater detail on what variables were evaluated for the regression models should be included. What was the primary and/or secondary outcome(s)? Trends over time for what? 

6. For the incidence rates, what was the denominator used? How well does this dataset represent the population in this region? As there are two health systems, are they both within the region and if yes, would they not be added to get the population estimate for the FNQ?

**Results**

-Does the analysis presented match the analysis plan?

-Are the results clearly and completely presented?

-Are the figures (Tables, Images) of sufficient quality for clarity?

Reviewer #1: Yes

Reviewer #2: Yes, clearly presented in the tables.

Reviewer #3: 1. The p-value for table 1 should be to one decimal place and afterwards to two (with the exception of <0.001). 

2. Were those who did not have ability to determine if there was cancer (~31) similar demographically to the cohort?

3. Is there an issue with the institutional ethics board with listing numbers less than 10 in Table 1? This can increase likelihood of having someone be identifiable and should be considered. I appreciate each person is described in a later table so as long as there are no concerns from the ethics board, that is okay.

4. Table 1 has odds ratios and typically is meant to be descriptive of the cohort. If this is reflecting the regression, perhaps it is best put as a separate table to make clear what the regression is looking at and the results.

5. Suggest changing the title to Association of TMP-SMX.

**Conclusions**

-Are the conclusions supported by the data presented?

-Are the limitations of analysis clearly described?

-Do the authors discuss how these data can be helpful to advance our understanding of the topic under study?

-Is public health relevance addressed?

Reviewer #1: Yes

Reviewer #2: Yes, the findings from this study further support active cancers and the treatment (chemotherapy) as significant risk factors for melioidosis. Bactrim prophylaxis to prevent melioidosis need to be further investigated to identify the high risk patients who may benefit from prophylaxis.

Reviewer #3: 1. In my opinion, the discussion is overstated. For example, they report NNT but cannot be certain that TMP-SMX is the only factor that 'prevented' an infection in the study. Given that only 11% of the cohort has a malignancy and that too highly variable in terms of type, duration and treatment, they are very underpowered to conclude that TMP-SMX is not helpful. While this may be true, the messaging needs to be changed and attenuated based on this study.

2. The changing incidence could be better explored by looking at data outside of their study cohort and the risk factors. Has this changing incidence been seen in other parts of australia or other countries with a high incidence? 

3. The statement of alcohol and smoking is not that relevant to this study and therefore should be again re-stated vs applying it to this cohort. Were there any other epidemiologic exposures that were considered?

4. I think much of the septra and immunosuppressive discussion can be reduced as this was not the specific goals of their study (beyond looking at septra and melioid risk/outcomes). 

5. The limitations paragaph needs to be improved. How did these limitations affect the study results and/or the interpretation of the study? What was done to mitigate these effects?

**Editorial and Data Presentation Modifications?**

Reviewer #1: (No Response)

Reviewer #2: Accept

Reviewer #3: (No Response)

**Summary and General Comments**

Reviewer #1: I enjoyed reading this manuscript.

The methodology is sound, the results are well presented, and the discussion regarding prophylactic use of TMP-SMX was excellent.

3 questions: 

1. Why did the authors use “died before discharge from melioidosis” (which could be phrased "died from melioidosis before discharge"), and not the total deaths from melioidosis irrespective of discharge? 

2. Is there a reference for the myelosuppressive grading scale used? If not, how confident are the authors of the accuracy of these subpopulations?

3. Partially out of interest, but with relevance to the paper, do the authors know the expected 12-month mortality for cancer patients with non-melioidosis infections in their region?

a. Similarly, is the 61% non-melioidosis-associated mortality expected or surprisingly high?

Reviewer #2: This is an important paper.

Reviewer #3: The authors present the results of a retrospective cohort study comparing outcomes of melioidosis in those with and without cancer. They identified that cancer of various types was present in 11% of cases and that there were some differences in characteristics. Those who received TMP were slightly less likely to develop infection but outcomes (death) were not significant different.

The methods are unclear at times and the analysis approaches should be made clearer.

The cohort is highly heterogeneous as it relates to cancer based on type, time (~2 decades), treatment, duration, stage etc and the impact of this on the study is not well considered.

The study is underpowered and this is not acknowledged by the investigators. Thus, much of the messaging in the discussion and conclusion needs to be tempered with this in mind.

PLOS authors have the option to publish the peer review history of their article (what does this mean?). If published, this will include your full peer review and any attached files.

Reviewer #1: No

Reviewer #2: Yes: Chee Yik Chang

Reviewer #3: No
---

## [Decision Letter · Decision Letter 1]

10 Sep 2024

Dear Dr. Hanson,

Thank you very much for submitting your manuscript "The characteristics and clinical course of patients with melioidosis and cancer" for consideration at PLOS Neglected Tropical Diseases. As with all papers reviewed by the journal, your manuscript was reviewed by members of the editorial board and by several independent reviewers. The reviewers appreciated the attention to an important topic. Based on the reviews, we are likely to accept this manuscript for publication, providing that you modify the manuscript according to the review recommendations. 

Thank you very much for responding in detail to the reviewers' concerns. While much of the manuscript has been improved, the final section focused on TMP-SMX treatment and development of melioidosis among patients with cancer remains problematic for the following reasons:

- As I understand it, this part of the analysis was conducted on patients at Cairns Hospital that were identified using the MOSAIQ programme. I suspect that these data were then integrated with the hospital laboratory culture data but I do not believe that this is explicitly stated. Please update the methods and the appropriate section of the results (starting at line 306) to ensure that readers understand both this strategy and that this is a separate parent cohort from the earlier analyses. 

- In this analysis, is it correct to assume is that most patients who were treated for cancer at Cairns Hospital are likely to have melioidosis diagnosed at the same hospital? 

- In order to appropriately evaluate the relative incidence of melioidosis in this cohort in those prescribed TMP-SMX vs not, (differential) follow up time is an essential consideration, yet does not seem to have been addressed. For example, if patients receiving TMP-SMX were followed up, on average, for longer time periods than patients not receiving TMP-SMX, conceivably more cases of melioidosis may be identified in those receiving TMP-SMX. Alternatively, if more patients receiving TMP-SMX died (due to their cancer or for reasons unrelated to melioidosis), then they would no longer be at risk for melioidosis and TMP-SMX may appear to be associated with lower rates of melioidosis. 

- I would caution the authors about using the phrase "impact of TMP-SMX" (or similar phrasing) when any relationship between TMP-SMX and melioidosis in this observational cohort can at best be considered a correlation, not causation.

- Confounders should also be considered in fully evaluating whether TMP-SMX is associated with altered incidence of melioidosis. For example, if TMP-SMX is preferentially prescribed to patients who are younger and have fewer comorbidities, then any reduced incidence of melioidosis may be related to the baseline characteristics of patients rather than TMP-SMX treatment. I recognize that the very low numbers of patients developing melioidosis (1 in the TMP-SMX group and 17 in the no-TMP-SMX group, as I read Table 4) essentially precludes any meaningful multivariable analysis. However, at the very least I would suggest including a table that describes the baseline characteristics of both groups to permit a qualitative evaluation by readers.

- The calculation of NNT is well-intentioned but, in my view, fraught with challenges when applied in this context. Please see Stang A, et al, J Clin Epidemiology 63 (2010) 820-25, for a summary of the issues. I would strongly consider eliminating NNT. Instead, I think that bolstering the analysis of melioidosis incidence among those receiving TMP-SMX vs not, as suggested above, will provide readers and clinicians with sufficient relevant data.

Sincerely,

T Eoin West, MD, MPH

Guest Editor

Ana LTO Nascimento

Section Editor

Thank you very much for responding in detail to the reviewers' concerns. While much of the manuscript has been improved, the final section focused on TMP-SMX treatment and development of melioidosis among patients with cancer remains problematic for the following reasons:

- As I understand it, this part of the analysis was conducted on patients at Cairns Hospital that were identified using the MOSAIQ programme. I suspect that these data were then integrated with the hospital laboratory culture data but I do not believe that this is explicitly stated. Please update the methods and the appropriate section of the results (starting at line 306) to ensure that readers understand both this strategy and that this is a separate parent cohort from the earlier analyses. 

- In this analysis, is it correct to assume is that most patients who were treated for cancer at Cairns Hospital are likely to have melioidosis diagnosed at the same hospital? 

- In order to appropriately evaluate the relative incidence of melioidosis in this cohort in those prescribed TMP-SMX vs not, (differential) follow up time is an essential consideration, yet does not seem to have been addressed. For example, if patients receiving TMP-SMX were followed up, on average, for longer time periods than patients not receiving TMP-SMX, conceivably more cases of melioidosis may be identified in those receiving TMP-SMX. Alternatively, if more patients receiving TMP-SMX died (due to their cancer or for reasons unrelated to melioidosis), then they would no longer be at risk for melioidosis and TMP-SMX may appear to be associated with lower rates of melioidosis. 

- I would caution the authors about using the phrase "impact of TMP-SMX" (or similar phrasing) when any relationship between TMP-SMX and melioidosis in this observational cohort can at best be considered a correlation, not causation.

- Confounders should also be considered in fully evaluating whether TMP-SMX is associated with altered incidence of melioidosis. For example, if TMP-SMX is preferentially prescribed to patients who are younger and have fewer comorbidities, then any reduced incidence of melioidosis may be related to the baseline characteristics of patients rather than TMP-SMX treatment. I recognize that the very low numbers of patients developing melioidosis (1 in the TMP-SMX group and 17 in the no-TMP-SMX group, as I read Table 4) essentially precludes any meaningful multivariable analysis. However, at the very least I would suggest including a table that describes the baseline characteristics of both groups to permit a qualitative evaluation by readers.

- The calculation of NNT is well-intentioned but, in my view, fraught with challenges when applied in this context. Please see Stang A, et al, J Clin Epidemiology 63 (2010) 820-25, for a summary of the issues. I would strongly consider eliminating NNT. Instead, I think that bolstering the analysis of melioidosis incidence among those receiving TMP-SMX vs not, as suggested above, will provide readers and clinicians with sufficient relevant data.

Reviewer's Responses to Questions

**Key Review Criteria Required for Acceptance?**

**Methods**

-Are the objectives of the study clearly articulated with a clear testable hypothesis stated?

-Is the study design appropriate to address the stated objectives?

-Is the population clearly described and appropriate for the hypothesis being tested?

-Is the sample size sufficient to ensure adequate power to address the hypothesis being tested?

-Were correct statistical analysis used to support conclusions?

-Are there concerns about ethical or regulatory requirements being met?

Reviewer #1: Yes

Reviewer #3: (No Response)

**Results**

-Does the analysis presented match the analysis plan?

-Are the results clearly and completely presented?

-Are the figures (Tables, Images) of sufficient quality for clarity?

Reviewer #1: Yes

Reviewer #3: (No Response)

**Conclusions**

-Are the conclusions supported by the data presented?

-Are the limitations of analysis clearly described?

-Do the authors discuss how these data can be helpful to advance our understanding of the topic under study?

-Is public health relevance addressed?

Reviewer #1: Yes

Reviewer #3: (No Response)

**Editorial and Data Presentation Modifications?**

Reviewer #1: Na

Reviewer #3: (No Response)

**Summary and General Comments**

Reviewer #1: The authors should be commended for their response to reviewer comments.

Reviewer #3: (No Response)

PLOS authors have the option to publish the peer review history of their article (what does this mean?). If published, this will include your full peer review and any attached files.

Reviewer #1: No

Reviewer #3: No

Figure Files:

Data Requirements:

Reproducibility:

References

---

## [Editor Report · Decision Letter 2]

14 Oct 2024

Dear Dr. Hanson,

We are pleased to inform you that your manuscript 'The characteristics and clinical course of patients with melioidosis and cancer' has been provisionally accepted for publication in PLOS Neglected Tropical Diseases.

Best regards,

T Eoin West, MD, MPH

Guest Editor

Ana LTO Nascimento

Section Editor

---

## [Editor Report · Acceptance letter]

18 Oct 2024

Dear Dr. Hanson,

We are delighted to inform you that your manuscript, "The characteristics and clinical course of patients with melioidosis and cancer," has been formally accepted for publication in PLOS Neglected Tropical Diseases.

Best regards,

Shaden Kamhawi

co-Editor-in-Chief

Paul Brindley

co-Editor-in-Chief
